# Malaria and dengue in Hodeidah city, Yemen: High proportion of febrile outpatients with dengue or malaria, but low proportion co-infected

**Rashad Abdul-Ghani**[1,2]\*, **Mohammed A. K. Mahdy**[1,2], **Sameer Alkubati**[3], **Abdullah A. Al-Mikhlafy**[4], **Abdullah Alhariri**[5], **Mrinalini Das**[6], **Kapilkumar Dave**[7], **Julita Gil-Cuesta**[8]

1 Department of Medical Parasitology, Faculty of Medicine and Health Sciences, Sana'a University, Sana'a, Yemen, 2 Tropical Disease Research Center, Faculty of Medicine and Health Sciences, University of Science and Technology, Sana'a, Yemen, 3 Department of Nursing, Faculty of Medicine and Health Sciences, Hodeidah University, Hodeidah, Yemen, 4 Department of Community Medicine, Faculty of Medicine and Health Sciences, University of Science and Technology, Sana'a, Yemen, 5 Preventive Medicine Unit, General Military Hospital, Hodeidah, Yemen, 6 Doctors without Borders, New Delhi, India, 7 Society for Education Welfare and Action—Rural, Jhagadia, India, 8 LuxOR Operational Research, Doctors without Borders, Brussels, Belgium

\* rashadqb@yahoo.com

**Data Availability Statement:** All relevant data are available within the paper as well as its Supporting Information files.

## Abstract

### Background

The emergence of dengue in malaria-endemic countries with limited diagnostic resources, such as Yemen, can be problematic because presumptive treatment of febrile cases as being malaria is a common practice. Co-infections with dengue and malaria are often over-looked and misdiagnosed as being a mono-infection because of clinical similarities. In Hodeidah city, Yemen, the capacity to conduct the diagnosis can be aggravated by the war context. To assess the magnitude of the problem, we determined the proportions of malaria, dengue and co-infection in relation to clinical characteristics among febrile outpatients.

### Methods

This cross-sectional study included 355 febrile outpatients from Hodeidah city during the malaria transmission season (September 2018 –February 2019). Sociodemographic and clinical characteristics were collected using a pre-designed, structured questionnaire. Malaria was confirmed using microscopy and rapid diagnostic tests (RDTs), while dengue was confirmed using RDTs.

### Results

Mono-infection proportions of 32.4% for falciparum malaria and 35.2% for dengue were found, where about two-thirds of dengue patients had a recent probable infection. However, co-infection with falciparum malaria and dengue was detected among 4.8% of cases. There was no statistically significant difference between having co-infection and mono-infection with malaria or dengue in relation to the sociodemographic characteristics. On the other

**Funding:** This study received funding from EMRO/TDR through the Joint EMRO/TDR Small Grants Scheme for Implementation Research in Communicable Diseases 2018-2019 (Grant ID: SGS18/37). However, the funders had no role in study design, data collection and analysis, decision to publish, or preparation of the manuscript.

**Competing interests:** The authors have declared that no competing interests exist.

hand, the odds of co-infection were significantly lower than the odds of malaria among patients presenting with sweating (OR = 0.1, 95% CI: 0.05–0.45; *p* <0.001), while the odds of co-infection were 3.5 times significantly higher than the odds of dengue among patients presenting with vomiting (OR = 3.5, 95% CI: 1.20–10.04; *p* <0.021). However, there were no statistically significant differences between having co-infection and mono-infection (malaria or dengue) in relation to other clinical characteristics.

## Conclusions

Mono-infection with malaria or dengue can be detected among about one-third of febrile out-patients in Hodeidah, while almost 5.0% of cases can be co-infected. Sociodemographic and clinical characteristics cannot easily distinguish malaria patients from dengue-infected or co-infected ones, reinforcing the necessity of laboratory confirmation and avoidance of treating febrile patients as being presumed malaria cases.

## Background

Malaria and dengue are major mosquito-borne diseases in terms of morbidity and mortality in tropical and subtropical countries. Globally, malaria was estimated to affect 229 million people and cause 409,000 deaths in 2019 across 87 countries [1]. Meanwhile, the global burden of dengue has been increasing over the past 50 years, with an annual incidence estimate of about 350 million infections, and about half of the world's population in endemic areas is at risk [2–4]. These diseases are often co-endemic and share similar clinical manifestations, with fever being the most common symptom [5]. Co-infections with these two diseases are often overlooked and misdiagnosed as being a mono-infection because of clinical similarities [6–8]. Malaria-dengue co-infection has been escalating after the increased reporting of dengue cases in malaria-endemic areas in various parts of the world since the first reported co-infected case in 2005 [9–14]. A systematic review reveals co-infection with both diseases in 20 countries in 2018 [5], ranging from 0.2% in Sierra Leone [15] to 23.0% among dengue-positive febrile patients in Pakistan [16].

In Yemen, over 165,000 confirmed malaria cases were reported by health facilities in 2019, predominantly caused by *P. falciparum* [1]. Concurrently, dengue cases have escalated; where several outbreaks caused by dengue virus (DENV) serotypes 2 and 3 have been reported between 2010 and 2012 [17–19]. DENV serotype 2 has been reported among about one-third of febrile patients with dengue-like illnesses in Hodeidah city, also called Al Hudaydah, in 2012 [19]. In Yemen, an increase of about six times in the number of suspected dengue cases was reported in 2016 compared with 2015 [20].

Diagnosis, care and control of malaria and dengue in Yemen have probably been affected by the unstable political situation and wars since 2012, where only half of the health facilities in the country were functional as of December 2018 [21]. A major problem is the lack of available diagnostic tests in Yemen that prompt physicians to assume that any acute febrile illness (AFI) is malaria and treat it as such, leading to unnecessary treatment of other AFIs. Inappropriate treatment of malaria-dengue co-infections can lead to severe complications or even death [14, 22–25], and unnecessary malaria treatment may contribute to the emergence and spread of drug resistance [26].

Misdiagnosis of febrile co-infections with shared clinical similarities is unavoidable in areas with overlapping endemicity. Although estimates of mono- and co-infections with malaria and dengue among febrile patients could be useful to clinicians, these are yet to be fully elucidated in Yemen. At the public health level, understanding the epidemiology of malaria and dengue, as mono- or co-infection, is essential for evidence-based approaches to appropriate control interventions. Therefore, we determined the proportions of malaria, dengue and malaria-dengue co-infection in relation to sociodemographic and clinical characteristics among febrile outpatients seeking healthcare and undergoing laboratory investigations for fever in the hospitals of Hodeidah city, west of Yemen—during malaria transmission season (November 2018 to April 2019).

## Subjects and methods

### Study design and setting

This hospital-based, cross-sectional study was conducted in accordance with STROBE guidelines [27] (S1 Checklist) in Hodeidah city in the period from November 2018 to April 2019. Hodeidah city is the capital of the governorate most afflicted by malaria in the country and comprises three districts: Al Mina, Al Hali and Al Hawak. It is located at the coordinates of 14˚ 48′08″N 42˚57′04″E and is a main port on the Red Sea in western Yemen. It is the second most populated city after Sana'a–the capital of Yemen, with a total population of 1,093,000 in 2017 [28]. It is endemic for malaria and has witnessed several dengue outbreaks with a recently reported increase in the number of suspected dengue cases.

### Study subjects

Febrile patients seeking healthcare in the outpatient departments of six tertiary care hospitals in Hodeidah were the target population of the study. Patients of any gender and age were included if referred for laboratory investigation of fever and having a directly observed axillary temperature of ≥37.5˚C at presentation provided that they gave informed consent and were residents of Hodeidah city for at least six months before the study. We excluded patients who or whose guardians refused to give informed consent. We defined co-infection as an infection with i) malaria based on microscopy and/or RDT and ii) recent probable dengue based on RDT [non-structural antigen 1 (NS1) and/or IgM-positive] on the same day of taking blood samples. However, past infection with dengue was defined by the detection of anti-DENV IgG alone.

### Sample size and sampling strategy

A minimum sample size of 273 was calculated using OpenEpi, version 3 (www.openepi.com) based on an expected co-infection proportion of 23% (the highest co-infection proportion found elsewhere [16]) at a confidence level of 95%, a precision of 5% and a design effect of 1.0. However, 355 patients were included from the three districts of Hodeidah city. Febrile patients referred to the laboratories of the hospitals of each district were invited to voluntarily participate in the study during the transmission season until the required sample size was attained.

### Data and sample collection

Laboratory technicians were trained on the study recruitment, informed consent and data collection and supervised by a co-investigator of the study. Data on sociodemographic characteristics (gender, age, educational status, employment status, temperature and clinical characteristics (sweating, chills, headache, muscle pain, joint pain and vomiting) were

systematically collected using a paper-based, pre-designed structured questionnaire in Arabic (S1 Table). Drops of blood were collected onto slides by finger-prick for preparing blood films and rapid testing of malaria. Then, about 2–3 ml of whole blood samples were collected into pre-labeled plain test tubes and left to clot at room temperature. Sera were then separated by centrifugation at 3000 rpm for five minutes for dengue rapid testing.

## Laboratory investigations

**Rapid diagnostic testing for malaria and dengue.** Blood drops were screened for malaria parasites using CareStart Malaria HRP2/pLDH (Pf/PAN) Combo RDTs to detect *P. falciparum* and non-falciparum species (AccessBio, New Jersey, USA). This test kit has been listed among the latest updated version of prequalified in vitro diagnostic products by the World Health Organization [29] and is one of the commonly used RDTs for malaria diagnosis in Yemen. Sera were tested for dengue through the detection of NS1 antigen and IgM/IgG antibodies with CareStart Dengue Combo RDTs. According to the guidelines of the US Centers for Disease Prevention and Control [30], the NS1 antigen is a useful tool for the diagnosis of acute dengue that can be detected in the serum as early as one day after the start of symptoms. Therefore, these RDTs can diagnose dengue at all clinical stages.

**Blood film microscopy.** Duplicate thick and thin blood films were prepared, stained with Giemsa for 20 minutes and examined under the oil-immersion lens of a light microscope by qualified microscopists according to standard procedures [31, 32].

## Data analysis

Data were double-entered and validated using EpiData software, version 3.1 (EpiData Association, Odense, Denmark) and transferred for analysis using Stata, version 16 (College Station, Texas, USA). Continuous variables were summarized as mean and standard deviation (SD) for normally distributed data or median and interquartile range (IQR) for non-normally distributed data, while categorical variables were summarized as frequencies and proportions. Differences in the presence of clinical characteristics between co-infection and each type of mono-infection were analyzed using the chi-square or Fisher's exact tests in bivariate analysis. Odds ratios (OR) for the difference between co-infection and each type of mono-infection were presented with their corresponding 95% confidence intervals (95% CI). Differences were considered statistically significant at $p$-values <0.05.

## Ethics statement

Ethical approval for this study was obtained from the Research Ethics Committee of the Faculty of Medicine and Health Sciences, University of Science and Technology (UST), Sana'a, Yemen (EAC/UST136). Additional approval was obtained from the Ethics Advisory Group (EAG) of The Union, Paris, France (EAG number: 14/19). Written or, in some cases, oral informed consent was obtained in Arabic from patients or their parents/guardians for children younger than 15 years. Those who gave oral consent expressed their cautious concerns about signing or finger-printing any documents and preferred to consent orally with complete anonymity, instead. However, the Research Ethics Committee of the UST approved obtaining oral consent in such situations. In addition, because these participants had also concerns about recording their voices, data collectors put their signatures on the corresponding consent forms with the day and date of oral consent after ensuring that the participants fully understood the objectives of the study and giving them the telephone number of the Principal Investigator for any queries.

## Results

### Characteristics of the study population

At the end of the study, 355 febrile patients were tested. Table 1 shows that the majority of patients were males (63.1%) and aged between 20 and 40 years (54.0%), with a median age of 28.0 ± 21.0 years and a mean temperature of 38.8 ± 0.7˚C. Most patients were unemployed (49.1%) and living in households with more than four members (64.5%). It also shows that febrile patients had a mean temperature of 38.8 ± 0.7˚C, with headache being the most frequent clinical feature (91.5%) followed by joint pain (82.3%), chills (66.2%) and sweating

**Table 1. Sociodemographic and clinical characteristics of febrile patients attending the outpatient departments in hospitals of Hodeidah city, Yemen (2018–2019)[*].**

| Characteristic | n | (%) |
|---|---:|---|
| **Gender** | | |
| Male | 224 | (63.1) |
| Female | 131 | (36.9) |
| **Age** (years) [a] | | |
| <20 | 92 | (26.1) |
| 20–40 | 190 | (54.0) |
| >40 | 70 | (19.9) |
| Median ± IQR: 28.0 ± 21.0 | | |
| **District** [b] | | |
| Al Mina | 129 | (37.8) |
| Al Hali | 119 | (34.9) |
| Al Hawak | 93 | (27.3) |
| **Education status** [c] | | |
| No formal education | 50 | (14.7) |
| Primary education | 75 | (22.0) |
| Secondary education or above | 216 | (63.3) |
| **Employment status** [d] | | |
| Unemployed | 142 | (49.1) |
| Public service employee | 51 | (17.7) |
| Private service employee | 96 | (33.2) |
| **Household size** (members) [e] | | |
| ≤4 | 102 | (35.5) |
| >4 | 185 | (64.5) |
| **Axillary temperature** (˚C) | | |
| Mean ± SD: 38.8 ± 0.7 | | |
| **Headache** | 325 | (91.5) |
| **Joint pain** | 292 | (82.3) |
| **Chills** | 235 | (66.2) |
| **Sweating** | 232 | (65.4) |
| **Muscle pain** | 102 | (28.7) |
| **Retro-orbital / ocular pain** | 96 | (27.0) |
| **Vomiting** | 72 | (20.3) |
| **Skin rash** | 3 | (0.8) |

[*] Total number of patients was 355

[**] *a* 3 missing cases; *b* 8 missing cases; *c* 14 missing or non-applicable cases; *d* 66 missing or non-applicable cases; *e* 68 missing cases; IQR, interquartile range.

(65.4%). In contrast, vomiting (20.3%) and skin rash (0.8%) were the least frequent clinical features among febrile outpatients.

## Proportions of malaria and dengue mono- and co-infections

Of 355 febrile patients, 32.4% had falciparum malaria as confirmed by microscopy and/or RDTs, where unmixed infection with *P. falciparum* was detected in 29.0% and 28.7% of patients by microscopy and RDTs, respectively. Microscopy revealed *P. vivax* in 3.1% of patients and mixed with *P. falciparum* in 1.1% of patients. On the other hand, 35.2% of patients were positive for dengue, where most cases were IgG-positive (13.0%) followed by those IgM/IgG-positive (9.6%) and NS1-positive (8.2%). Approximately two-thirds of dengue-positive patients had recent probable dengue, while 36.8% had past infections. Co-infection with falciparum malaria and recent probable dengue was detected among 4.8% of patients (Table 2).

## Comparison between co-infection and mono-infection with malaria and dengue in relation to sociodemographic and clinical characteristics

Table 3 shows that there was no statistically significant difference between co-infection and mono-infection with malaria or dengue and the male gender, age of twenty years or older, having secondary education or above, being unemployed or living within households of more

**Table 2. Positivity of malaria and dengue among febrile patients attending the outpatient departments in hospitals of Hodeidah city, Yemen (2018–2019)[*].**

| Infection status | | *n* | (%) |
|---|---|---|---|
| **Microscopy-confirmed malaria** | | | |
| | P. falciparum | 103 | (29.0) |
| | *P. vivax* | 11 | (3.1) |
| | Co-infection with *P. falciparum* and *P. vivax* | 4 | (1.1) |
| **RDT-confirmed malaria** | | | |
| | *P. falciparum* | 102 | (28.7) |
| | Non-falciparum species | 12 | (3.4) |
| | Falciparum and non-falciparum species | 9 | (2.5) |
| **Total confirmed falciparum malaria** (microscopy and/or RDT) | | **115** | **(32.4)** |
| **Dengue RDT result** | | | |
| | IgM-positive | 11 | (3.1) |
| | IgG-positive | 46 | (13.0) |
| | IgM- and IgG-positive | 34 | (9.6) |
| | NS1-positive | 29 | (8.2) |
| | NS1- and IgM- and/or IgG-positive | 5 | (1.4) |
| **Total confirmed dengue** (RDT) | | **125** | **(35.2)** |
| **Categories of RDT-confirmed dengue[a]** | | | |
| | Recent probable (positive for IgM and/or NS1 irrespective of IgG) | 79 | (63.2) |
| | Past (positive for IgG only) | 46 | (36.8) |
| **Malaria-dengue co-infection** [b] | | **17** | **(4.8)** |

[*] Total number of patients was 355; RDT, rapid diagnostic test; IgM, immunoglobulin M; IgG, immunoglobulin G; NS1, Non-structural protein 1; IQR, interquartile range

[a] Calculated from dengue-positive cases

[b] Cases co-infected with falciparum malaria and recent probable dengue (because dengue was not diagnosed in patients with vivax malaria).

**Table 3. Comparison between co-infection and mono-infection with malaria and dengue among febrile patients from Hodeidah city of Yemen in relation to certain sociodemographic and clinical characteristics (2018–2019).**

| Characteristics* | Malaria N = 98 | | Dengue N = 108 | | Co-infection N = 17 | | Co-infection *vs.* malaria | | | Co-infection *vs.* dengue | | |
|---|---|---|---|---|---|---|---|---|---|---|---|---|
| | *n* | (%) | *n* | (%) | *n* | (%) | OR | (95% CI) | *p*-value** | OR | (95% CI) | *p*-value** |
| Male gender | 65 | (66.3) | 65 | (60.2) | 14 | (82.4) | 1.7 | (0.63–8.97) | 0.188 | 3.1 | (0.82–11.64) | 0.078 |
| Age of 20 years or older | 74 | (75.5) | 76 | (70.4) | 14 | (82.4) | 0.7 | (0.19–2.74) | 0.760 | 0.5 | (0.15–2.03) | 0.401 |
| Secondary education or above | 63 | (64.3) | 60 | (55.6) | 10 | (58.8) | 0.9 | (0.31–3.12) | 0.862 | 0.7 | (0.21–2.09) | 0.671 |
| Unemployment | 43 | (43.9) | 47 | (43.5) | 6 | (35.3) | 1.2 | (0.39–3.55) | 0.777 | 1.4 | (0.49–4.10) | 0.604 |
| Living in a household of ≥ 4 members | 53 | (54.1) | 62 | (57.4) | 13 | (76.5) | 0.4 | (0.11–1.19) | 0.113 | 0.4 | (0.13–1.35) | 0.185 |
| Sweating | 77 | (78.6) | 59 | (54.6) | 6 | (35.3) | 0.1 | (0.05–0.45) | <0.001 | 0.5 | (0.16–1.32) | 0.145 |
| Chills | 75 | (76.5) | 65 | (60.2) | 14 | (82.4) | 1.4 | (0.38–5.42) | 0.598 | 3.1 | (0.84–11.37) | 0.091 |
| Headache | 90 | (91.8) | 98 | (90.7) | 15 | (88.2) | 0.7 | (0.13–3.45) | 0.629 | 0.8 | (0.15–3.84) | 0.745 |
| Muscle pain | 25 | (25.5) | 31 | (28.7) | 6 | (35.3) | 1.6 | (0.53–4.75) | 0.404 | 1.4 | (0.46–4.00) | 0.581 |
| Joint pain | 81 | (82.7) | 97 | (89.8) | 13 | (76.5) | 0.7 | (0.53–4.79) | 0.542 | 0.4 | (0.10–1.35) | 0.116 |
| Retro-orbital / ocular pain | 20 | (20.4) | 49 | (45.4) | 5 | (29.4) | 1.6 | (0.51–5.19) | 0.406 | 0.5 | (0.16–1.54) | 0.217 |
| Vomiting | 33 | (33.7) | 22 | (20.4) | 8 | (47.1) | 1.8 | (0.62–5.00) | 0.291 | 3.5 | (1.20–10.04) | 0.021 |

*Skin rash was excluded from analysis because it was detected among two malaria and two dengue cases but among none of the co-infected cases.

**p-value for bivariate analysis of characteristics between co-infection and each type of mono-infection; OR, odds ratio; CI, confidence interval.

than four members. On the other hand, the odds of co-infection were significantly lower than the odds of malaria among patients presenting with sweating (OR = 0.1, 95% CI: 0.05–0.45; *p* <0.001), while the odds of co-infection were 3.5 times significantly higher than the odds of dengue among patients presenting with vomiting (OR = 3.5, 95% CI: 1.20–10.04; *p* <0.021). In contrast, there were no statistically significant differences between co-infection and mono-infection with either type among febrile patients concerning other studied clinical features.

## Discussion

The proportions of mono-infection with falciparum malaria and dengue among febrile patients seeking healthcare in Hodeidah city were comparable, where each type of infection was diagnosed among about one-third of patients. Meanwhile, about two-thirds of dengue cases were recent probable infections, which is higher than that reported for acute dengue among about one-third of febrile patients with dengue-like illnesses in Hodeidah in 2012 using enzyme-linked immunosorbent assay (ELISA) and polymerase chain reaction (PCR) [19]. The high mono-infection proportions beside the high proportion of febrile cases negative for both types of infection underscore the necessity of laboratory confirmation and avoiding treatment of AFIs as being presumed malaria cases. Moreover, the emergence of other AFIs in the country, such as chikungunya [33], warrants further investigations and broadening the diagnostic panel used for passive case detection of AFIs.

Malaria and dengue co-existence was first reported among prisoners during a febrile outbreak in Hodeidah in mid-2018 [34] but without co-infection estimates. The present study unveiled a co-infection proportion of as low as almost 5%, indicating that malaria infections are more frequent among febrile patients having no dengue than dengue-infected ones. This finding is in line with a conclusion of a recent meta-analysis [35] that the odds of malaria are significantly lower in dengue-infected patients compared with non-infected ones. Such uncommon co-infection with malaria and dengue could be attributed to several factors, including different vectors and their habitats. Nevertheless, co-infection should not be ignored once one type of infection is diagnosed.

The co-infection proportion in the present study is lower than that (30.4%) recently reported among febrile patients from Hodeidah [36]. However, unlike the recruitment of outpatients resident in Hodeidah city for at least six months before the study, the latter study [36] recruited outpatients and inpatients from rural and urban areas, where 51.4% of co-infections were diagnosed among inpatients compared with 16.5% among outpatients. The role of using different dengue diagnostic techniques in the difference between the co-infection proportions among outpatients in the two studies could not be ruled out, where NS1 antigen and IgM/IgG antibodies were detected using RDTs in the present study whereas IgM/IgG antibodies were detected using ELISA in the previous study.

Malaria diagnosis in the present study using microscopy and antigen detection using RDTs may underestimate the proportion of falciparum malaria cases among febrile patients, which can lead to inadequate detection of malaria as a mono- or co-infection as shown by recently published literature [37–40]. Therefore, the findings of this study should be interpreted cautiously in this context, and the underestimation of malaria in mono- and co-infected febrile patients could not be ruled out. On the other hand, NS1 helps detect recent infections as early as one day from the onset of symptoms when combined with IgM and IgG antibodies for dengue diagnosis [30], allowing to cover all clinical stages of the disease. Although the diagnostic accuracy of RDTs to detect NS1 antigen is yet to be evaluated, the capacity of such RDTs to capture some PCR-negative cases at the end of the viremic phase has been evidenced among febrile patients [41]. Given that the RDT used in the present study might not be sensitive enough to enable the detection of all dengue infections, the large gap between the proportion of dengue mono-infection and malaria-dengue co-infection in a random sample of febrile patients could still be informative. RDTs are useful tools to screen for dengue in limited-resource countries with limited or unavailable reference diagnostic services [42], but it is recommended to use ELISA to confirm dengue diagnosis in suspected cases in such countries. However, IgM ELISAs can be less than 50% sensitive for confirming primary infection for at least 4 days due to the delayed increases in antibody titers [42]. Overall, the underestimation of mono- or co-infection with malaria and dengue among febrile patients in the present study could not be precluded. Such underestimation is not only due to the inadequacy of the detection methods used that have lower levels of sensitivity and specificity compared to molecular methods but also due to several other factors such as disease seasonality and the level of overlapping endemicity. Meanwhile, presumptive diagnosis of febrile cases by Yemeni physicians as dengue based on thrombocytopenia or malaria based on symptoms in a malaria-endemic area can lead to an overestimation of the proportion of either type of infection among patients with AFIs. This situation is typical of developing countries with limited diagnostic resources, where the actual incidence of infection with malaria and/or dengue is not correctly detected [14]. Therefore, the development of novel diagnostic tools, preferably as a single format, for both types of infection has been suggested [14].

Compared with malaria-dengue co-infection in Hodeidah and with the use of similar diagnostic tools, a higher proportion (7.2%; 27/367) was reported among febrile patients from Odisha state of India during a dengue outbreak in 2011 [23], but a lower proportion (0.2%; 3/1260) was reported among febrile patients from Sierra Leone in 2012–2013 [15]. The variations in co-infection proportions could be attributed to several factors, including the endemicity levels of the two diseases and the mobility patterns of populations in endemic countries. Unlike the present study, ELISA was the most frequently used method for the detection of dengue NS1, IgM and/or IgG followed by detecting the virus nucleic acid by PCR in other regions of the world [11–13, 16, 43–57]. Therefore, the differences in detection methods and study population categories make it inappropriate to compare the co-infection proportion among febrile patients from Hodeidah with those reported in these studies. Against this background,

higher co-infection with microscopy-confirmed malaria and ELISA-confirmed dengue was reported among 23.0% of dengue-positive febrile patients in Karachi, Pakistan in 2007–2008 [16], but lower proportions were reported from Punjab province of Pakistan (1.1% in 2003–2004 and 2.0% in 2012) for co-infection with microscopy-confirmed *Plasmodium* species and ELISA-confirmed dengue [13, 44]. Based on malaria microscopy and dengue ELISA, lower co-infection proportions (0.3–3.0%) were reported among febrile Indian patients from different Indian states between 2005 and 2014 [12, 48–50, 52, 55, 56], but higher co-infection proportions were reported among febrile patients from Mumbai during monsoons in 2014 and 2015, being 10.3% and 6.7%, respectively [54]. In Africa, malaria-dengue co-infection was reported among 2.0–6.0% of febrile Nigerian patients between 2008 and 2016 [43, 45, 53, 57], 3.0% (7/218) of Ghanaian patients with confirmed malaria [51] and 8.5% of febrile Tanzanian patients in 2013 [46]. In South America, however, lower co-infection proportions ranging from 1.0% (17/1723) to 2.8% (44/1578) were reported among febrile patients from French Guiana and the Brazilian Amazon [11, 47].

The present study revealed that sociodemographic characteristics could not be useful predictors to differentiate malaria-dengue co-infection from either type of mono-infection. Such finding is consistent with that reported among in-patients and out-patients in the city [36]. It is to be noted that gender bias might be introduced into the present study due to the recruitment of more males (63.0%), which should be considered in future studies to test the significance of difference between co-infection and mono-infection with malaria or dengue in relation to gender. Gender can be associated with differences in the likelihood of exposure to dengue vectors [58, 59].

Clinical features could not easily distinguish co-infection from either type of infection, where the quite similar clinical manifestations of malaria and dengue usually lead to ignoring the co-infection among febrile patients once a mono-infection with either type is confirmed. Yet, febrile patients presenting with sweating were significantly more likely to be infected with malaria than being co-infected, while those presenting with vomiting were significantly more likely to be co-infected than being infected with dengue. In another context, vomiting was found to be significantly more frequent among febrile patients co-infected with dengue and vivax malaria than those mono-infected with dengue in the Brazilian Amazon [47]. The absence of differences between co-infection and either type of mono-infection regarding other clinical features, such as joint pain and retro-orbital pain, makes it difficult to predict whether febrile patients confirmed with malaria might be co-infected with dengue and vice versa. Therefore, diagnosis of one type of infection should not exclude the presence of the other whose treatment could be ignored or, at least, delayed. In line with this, Abbasi et al. [16] and Mohapatra et al. [23] found a clinical similarity between malaria-dengue co-infection and dengue among Pakistani and Indian febrile patients, respectively. Because the present study was limited to outpatients not presenting with severe disease or complications due to the difficulty in accessing inpatients, the association of co-infection with disease severity needs to be investigated. In this context, more severe clinical presentations were found in co-infected patients admitted to hospitals compared with those mono-infected with either type of disease in French Guiana [22].

The outcomes of this study will contribute to informing healthcare personnel about malaria and dengue mono- and co-infection proportions among patients with AFIs in Hodeidah city. This is particularly important because treating febrile patients as presumed malaria cases is a common practice where diagnostic services are deficient or lacking, leading to unnecessary and irrational antimalarial treatment of AFIs other than malaria. On the other hand, confirming malaria or dengue mono-infection in co-infected patients may lead to ignoring the supportive treatment of dengue or treatment of malaria, respectively. Although no specific

medication for dengue currently exists, supportive treatment with antipyretics, analgesics and fluid replacement in case of dehydration is recommended and can be life-saving [60]. Another common practice by Yemeni physicians is using platelet count as an indicator of dengue among patients with AFIs wherever dengue diagnostics are not easily accessible or available. Despite being, among others, an indicator of dengue [61], thrombocytopenia can also be present in malaria [62, 63], necessitating its laboratory confirmation in febrile patients. At the public health level, the outcomes of the present study can help the National Malaria Control Programme (NMCP) in orienting its current case management recommendations. In this respect, the high proportion of dengue among febrile patients in Hodeidah can lead to their presumptive treatment as being malaria, confronting with the strategic component of malaria case management adopted by the NMCP that conforms to the WHO recommendation of universal diagnostic testing before malaria treatment [64, 65]. Therefore, educational and training interventions should be tailored and implemented by the NMCP to increase the adherence of public and private sector physicians to the national malaria treatment guidelines, including the treatment of confirmed rather than presumed malaria cases. Because the NMCP is the authority responsible for dengue control in the country, implementation of interventions for assessing the diagnostic accuracy of RDTs for dengue diagnosis and broadening the extent of their availability in malaria-endemic areas can help translate its national malaria treatment guidelines into practice.

Apart from studying mono- or co-infection with malaria and dengue among febrile patients, the potential co-infection with other AFIs with similar differential diagnosis but not investigated in the present study should be considered. This is evidenced by the finding that about one-third of febrile patients seeking healthcare in Hodeidah tested negative for both malaria and dengue. In this context, the emergence of a chikungunya outbreak was reported in Hodeidah city in 2011 [33] followed by reporting the co-circulation of chikungunya virus with DENV in the city in 2012 [19], raising the potential co-infection with other infectious causes of AFIs. Having the same vector and sharing similar symptoms with dengue, chikungunya should also be given priority when investigating AFIs in patients living in Hodeidah, preferably after assessing its burden as mono- and co-infection. In many malaria-endemic areas in low- and middle-income countries, AFIs other than malaria and dengue, such as chikungunya, leptospirosis and Q-fever, are usually neglected with the lack of a comprehensive, standardized and multi-center etiology research on the infectious causes of fever, even for severe cases [66]. Studies on non-malarial infectious causes of AFIs, either as mono- or co-infection with one or more circulating agents, are critically needed to raise the awareness of clinicians and disease control programmes about the local epidemiology and age distribution of the causes of AFIs as well as the burden of clinically indistinguishable febrile co-infections. A better understanding of the local epidemiology of malaria co-infections besides the non-malarial causes of AFIs using robust immunological and molecular techniques is critical for guiding the diagnostic approaches to AFIs and improving the outcomes of clinical management.

This study is limited by the fact that it was hospital-based recruiting symptomatic patients without severe infections due to difficulties in accessibility to inpatients. Therefore, its findings may not be generalizable and do not reflect the epidemiologic status of co-infection, neither among healthcare-seeking patients nor at the community level. However, the findings of the present study can be a basis for further large-scale hospital- and community-based studies in areas of the country co-endemic for both types of infection. Another limitation was introduced by the low co-infection proportion, which might make the study not powered enough to compare co-infection with malaria or dengue mono-infection in relation to clinical characteristics. Accordingly, more extensive studies should build on the findings of the present study to further assess such differences. On the other hand, it is recommended to assess the outcomes and

complications of malaria and dengue, as either mono- or co-infection, through longitudinal studies on sub-sets of patients.

Although dengue was diagnosed using RDTs, such types of tests also detect NS1 antigen as a useful marker of recent or acute dengue [29] together with IgM and IgG antibodies. However, the possible underestimation of dengue mono- or co-infection attributed to the nature of tests used in this study warrants further research adopting more robust immunological, such as IgM ELISA, and molecular techniques for dengue diagnosis together with the evaluation of the diagnostic accuracy of RDTs to broaden their accessibility and affordability as diagnostic tools in such a resource-limited country.

In conclusion, the proportions of mono-infection with malaria and dengue are comparable among about one-third of febrile outpatients in Hodeidah, while almost 5% of febrile cases can be co-infected. Sociodemographic characteristics and clinical features cannot easily distinguish malaria patients from dengue-infected or co-infected ones, reinforcing the necessity of proper laboratory diagnosis and avoidance of treating febrile patients as being presumed malaria cases in such a setting. Further large-scale studies are recommended to assess the burden of malaria and dengue, as mono- or co-infections, among febrile patients using more robust diagnostic tools in areas of the country co-endemic for both types of infection. Moreover, assessment of the epidemiologic burden of other infectious causes of AFIs, particularly chikungunya, as a mono- or co-infection is warranted.

## Supporting information

**S1 Checklist. STROBE checklist.**
(DOC)

**S1 Table. Questionnaire (English & Arabic versions).**
(DOCX)

**S2 Table. Data set.**
(XLS)

## Acknowledgments

We thank the patients who agreed to participate in the study and the laboratory technicians who assisted while conducting the study. We also thank Dr. Adel Al-Jasari, Malaria and Vector Control Officer, WHO Country Office in Sana'a, Yemen for his critical comments and feedback during manuscript preparation. This research was conducted through the Structured Operational Research and Training Initiative (SORT IT), a global partnership led by the Special Program for Research and Training in Tropical Diseases at the World Health Organization (WHO/TDR). The model is based on a course developed jointly by the International Union against Tuberculosis and Lung Disease (The Union) and Medécins sans Frontières (MSF/Doctors Without Borders). The specific SORT IT program which resulted in this publication was jointly developed and implemented by: The Union Southeast Asia Office, New Delhi, India; the Centre for Operational Research, The Union, Paris, France; The Union, Mandalay, Myanmar; The Union, Harare, Zimbabwe; MSF Luxembourg Operational Research (LuxOR); MSF Operational Center Brussels (MSF OCB); Jawaharlal Institute of Postgraduate Medical Education and Research (JIPMER), Puducherry, India; Post Graduate Institute of Medical Education and Research (PGIMER), Chandigarh, India; All India Institute of Medical Sciences (AIIMS), New Delhi, India; ICMR- National Institute of Epidemiology, Chennai, India; Society for Education Welfare and Action (SEWA)–Rural, Jhagadia, India; Common Management Unit (AIDS, TB & Malaria), Ministry of National Health Services, Regulations and Coordination,

Islamabad, Pakistan; and Kidu Mobile Medical Unit, His Majesty's People's Project and Jigme Dorji Wangchuck National Referral Hospital, Thimphu, Bhutan.

## Author Contributions

**Conceptualization:** Rashad Abdul-Ghani, Mohammed A. K. Mahdy.

**Data curation:** Rashad Abdul-Ghani, Mohammed A. K. Mahdy, Abdullah Alhariri.

**Formal analysis:** Rashad Abdul-Ghani.

**Funding acquisition:** Rashad Abdul-Ghani, Mohammed A. K. Mahdy.

**Investigation:** Sameer Alkubati.

**Methodology:** Rashad Abdul-Ghani, Sameer Alkubati.

**Project administration:** Sameer Alkubati, Abdullah A. Al-Mikhlafy.

**Resources:** Mrinalini Das, Kapilkumar Dave, Julita Gil-Cuesta.

**Software:** Mrinalini Das, Kapilkumar Dave, Julita Gil-Cuesta.

**Supervision:** Rashad Abdul-Ghani, Mohammed A. K. Mahdy, Sameer Alkubati, Abdullah A. Al-Mikhlafy, Abdullah Alhariri, Mrinalini Das, Kapilkumar Dave, Julita Gil-Cuesta.

**Validation:** Rashad Abdul-Ghani, Mrinalini Das, Kapilkumar Dave, Julita Gil-Cuesta.

**Writing – original draft:** Rashad Abdul-Ghani, Mrinalini Das, Kapilkumar Dave, Julita Gil-Cuesta.

**Writing – review & editing:** Rashad Abdul-Ghani, Mohammed A. K. Mahdy, Sameer Alkubati, Abdullah A. Al-Mikhlafy, Abdullah Alhariri, Mrinalini Das, Kapilkumar Dave, Julita Gil-Cuesta.

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
