## [Decision Letter · Decision Letter 0]

3 Mar 2021

PONE-D-21-03805

Malaria and dengue in Hodeidah city, Yemen: one-third of febrile outpatients with dengue or malaria but low proportion co-infected

PLOS ONE

Dear Dr. Abdul-Ghani,

Thank you for submitting your manuscript to PLoS ONE. After careful consideration, we felt that your manuscript requires revision, following which it can possibly be reconsidered. Although your manuscript was of interest to the reviewer, major concerns were related to study design and data interpretation. A major concern was related to the method for malaria diagnosis that seems not adequate for the detection of low levels of malaria infection, and co-infections. According to the reviewers, since most of the data were based on RDT and microscopy, there is a likelihood of underrepresentation of co-infections and other species of malaria. In addition, the reviewer complains that RDTs can result in false positives in dengue when malaria is positive, and an IgM Dengue ELISA to confirm these would be more useful while generalizing the results to other non-resource limited countries / institutions. In addition, a significant number of issues should be clarified and/or adjust otherwise the MS’s results may be compromised. For your guidance, a copy of the reviewers' comments was included below .

We look forward to receiving your revised manuscript.

Kind regards,

Luzia Helena Carvalho, Ph.D.

Academic Editor

PLOS ONE

Journal Requirements:

2.  We note that Figure 1 in your submission contain map images which may be copyrighted. All PLOS content is published under the Creative Commons Attribution License (CC BY 4.0), which means that the manuscript, images, and Supporting Information files will be freely available online, and any third party is permitted to access, download, copy, distribute, and use these materials in any way, even commercially, with proper attribution. For these reasons, we cannot publish previously copyrighted maps or satellite images created using proprietary data, such as Google software (Google Maps, Street View, and Earth). For more information, see our copyright guidelines: http://journals.plos.org/plosone/s/licenses-and-copyright.

2.1. You may seek permission from the original copyright holder of Figure 1 to publish the content specifically under the CC BY 4.0 license. 

2.2. If you are unable to obtain permission from the original copyright holder to publish these figures under the CC BY 4.0 license or if the copyright holder’s requirements are incompatible with the CC BY 4.0 license, please either i) remove the figure or ii) supply a replacement figure that complies with the CC BY 4.0 license. Please check copyright information on all replacement figures and update the figure caption with source information. If applicable, please specify in the figure caption text when a figure is similar but not identical to the original image and is therefore for illustrative purposes only.

3. We note that some participants gave oral consent. In the Methods, please state the following:

- Why written consent could not be obtained in some cases

- Whether the Institutional Review Board (IRB) approved use of oral consent

- How oral consent was documented

For more information, please see our guidelines for human subjects research: https://journals.plos.org/plosone/s/submission-guidelines#loc-human-subjects-research

4. Please include additional information regarding the survey or questionnaire used in the study and ensure that you have provided sufficient details that others could replicate the analyses. For instance, if you developed a questionnaire as part of this study and it is not under a copyright more restrictive than CC-BY, please include a copy, in both the original language and English, as Supporting Information.

"The funders had no role in study design, data collection and analysis, decision to publish, or preparation of the manuscript"

6. We note that you have indicated that data from this study are available upon request. PLOS only allows data to be available upon request if there are legal or ethical restrictions on sharing data publicly. For information on unacceptable data access restrictions, please see http://journals.plos.org/plosone/s/data-availability#loc-unacceptable-data-access-restrictions.

Reviewers' comments:

Reviewer's Responses to Questions

**Comments to the Author**

1. Is the manuscript technically sound, and do the data support the conclusions?

Reviewer #1: Yes

Reviewer #2: Partly

2. Has the statistical analysis been performed appropriately and rigorously? 

Reviewer #1: Yes

Reviewer #2: Yes

3. Have the authors made all data underlying the findings in their manuscript fully available?

Reviewer #1: Yes

Reviewer #2: Yes

4. Is the manuscript presented in an intelligible fashion and written in standard English?

Reviewer #1: Yes

Reviewer #2: Yes

5. Review Comments to the Author

Reviewer #1: 1. The article is relevant to only those practicing in Yemen as these findings cannot be generalized to other countries and populations. Hence publishing these results in local journals for local information and dissipation of knowledge would be much more useful.

2. The choice of only OPD patients with co-infection defeats the purpose of the study as the fact that dengue or malaria can be missed and thus result in complications to the patient is what is important in such co-infection.

3. The final outcomes of the patients has not been analysed which could add a lot of value to the study

4. The RDT s can show false positive tests in dengue when malaria is positive, and an IgM Dengue ELISA to confirm these would be more useful while generalizing the results to other non-resource limited countries / institutions.

Reviewer #2: The authors have carried out a systematic study to assess the co-infections of malaria and dengue in Hodeidah city during the malaria transmission season (September 2018 – February 2019). The study has been conducted systematically. However, the interpretation and discussion of the study is incomplete. Recently literature is also not appropriately cited. Based on these points, the following suggestions are made to improve this study:

(i) There is a biased sex ratio of 63% males in the recruited patients in this study. This should be appropriately addressed in the discussion and the possible effect of this bias on the outcomes of this study should be discussed.

(ii) The study has used RDT, and microscopy as methods of diagnosis for malaria. This is a partially correct methodology since ample evidence exists in literature that RDT and microscopy are not adequate for the detection of malaria- and co-infections are especially hard to detect. There should be a detailed discussion on this topic with an emphasis on recently published reports such as Mandage R, et al. Emerg Infect Dis. 2020; Kaur, C., et. al., BMC Res Notes 2020; Watson OJ, et al. BMJ Glob Health. 2019; Anstey NM, Grigg MJ. J Infect Dis. 2019; Berhane A et. al., Emerg Infect Dis. 2018; to name a few.

(iii) Since most of the data here is generated based only on RDT and microscopy, there is a likelihood of underrepresentation of co-infections and other species of malaria. This is incompletely discussed from lines 218-226. This should be further discussed to include all possible reasons for the underdiagnosis of malaria and dengue including but not limited to seasonality, the nature of tests used, the relative sensitivity and specificities of these tests etc.

(iv) The authors have mentioned the demographics on unemployment status and crowding in the results section (line 167). The impact of these parameters on malaria and dengue infection/ transmission should be discussed since this has been mentioned in the result.

(v) The purpose of Fig. 1 is unclear- including a discussion on the geography of the Hodeidah city and its impact on malaria and dengue transmission and incidence, might warrant inclusion of this figure. Otherwise, it may be removed.

Minor comment: Spelling error in line 91.

6. PLOS authors have the option to publish the peer review history of their article (what does this mean?). If published, this will include your full peer review and any attached files.

Reviewer #1: No

Reviewer #2: No

---

## [Author Response · Author response to Decision Letter 0]

28 Apr 2021

Response to Reviewers’ Comments

Dear Editor,

Thank you for giving us the opportunity to improve the quality of our manuscript. Our thanks are also due to the Reviewers for their helpful comments and the raised issues and concerns. Changes in the revised manuscript are made with track changes. The line numbers mentioned below also referred to the lines in the revised manuscript with track changes. Detailed responses to the comments raised are as follows:

Responses to the comments by the Academic Editor:

Comment: “Thank you for submitting your manuscript to PLoS ONE. After careful consideration, we felt that your manuscript requires revision, following which it can possibly be reconsidered. Although your manuscript was of interest to the reviewer, major concerns were related to study design and data interpretation. A major concern was related to the method for malaria diagnosis that seems not adequate for the detection of low levels of malaria infection, and co-infections. According to the reviewers, since most of the data were based on RDT and microscopy, there is a likelihood of underrepresentation of co-infections and other species of malaria. In addition, the reviewer complains that RDTs can result in false positives in dengue when malaria is positive, and an IgM Dengue ELISA to confirm these would be more useful while generalizing the results to other non-resource limited countries / institutions. In addition, a significant number of issues should be clarified and/or adjust otherwise the MS’s results may be compromised. For your guidance, a copy of the reviewers' comments was included below.”

Response: We thank you and the Reviewers for the invaluable comments that improved the quality of our manuscript. We addressed the comments raised in the revised version of the manuscript. The limitations of the techniques used in the study were discussed in the revised manuscript, highlighting the importance of cautious interpretation of the findings within their context and recommending further large-scale studies adopting more robust immunological and molecular techniques.

Responses to PLOS ONE requirements:

Comment: “1. Please ensure that your manuscript meets PLOS ONE's style requirements, including those for file naming. The PLOS ONE style templates can be found at

https://journals.plos.org/plosone/s/file?id=ba62/PLOSOne_formatting_sample_title_authors_affiliations.pdf.”

Response: We revised the manuscript according to PLOS ONE’s style requirements (Kindly see the revised manuscript). 

Comment: “2. We note that Figure 1 in your submission contain map images which may be copyrighted. All PLOS content is published under the Creative Commons Attribution License (CC BY 4.0), which means that the manuscript, images, and Supporting Information files will be freely available online, and any third party is permitted to access, download, copy, distribute, and use these materials in any way, even commercially, with proper attribution. For these reasons, we cannot publish previously copyrighted maps or satellite images created using proprietary data, such as Google software (Google Maps, Street View, and Earth). For more information, see our copyright guidelines: http://journals.plos.org/plosone/s/licenses-and-copyright.

2.1. You may seek permission from the original copyright holder of Figure 1 to publish the content specifically under the CC BY 4.0 license. 

2.2. If you are unable to obtain permission from the original copyright holder to publish these figures under the CC BY 4.0 license or if the copyright holder’s requirements are incompatible with the CC BY 4.0 license, please either i) remove the figure or ii) supply a replacement figure that complies with the CC BY 4.0 license. Please check copyright information on all replacement figures and update the figure caption with source information. If applicable, please specify in the figure caption text when a figure is similar but not identical to the original image and is therefore for illustrative purposes only.

Natural Earth (public domain): http://www.naturalearthdata.com/.”

Response: We preferred to delete the figure as indicated by Reviewer #2 and according to your comment. 

Comment: “3. We note that some participants gave oral consent. In the Methods, please state the following:

- Why written consent could not be obtained in some cases

- Whether the Institutional Review Board (IRB) approved use of oral consent

- How oral consent was documented

For more information, please see our guidelines for human subjects research: https://journals.plos.org/plosone/s/submission-guidelines#loc-human-subjects-research.”

Response: We stated the reason for not obtaining written consent from some participants, the approval of this action by the Research Ethics Committee and the documentation of such oral consent (Lines 164-170). 

Comment: “4. Please include additional information regarding the survey or questionnaire used in the study and ensure that you have provided sufficient details that others could replicate the analyses. For instance, if you developed a questionnaire as part of this study and it is not under a copyright more restrictive than CC-BY, please include a copy, in both the original language and English, as Supporting Information.”

Response: S2 Table (English and Arabic versions of the questionnaire) was included as Supporting Information.

Comment: “5. Thank you for stating the following financial disclosure:

"The funders had no role in study design, data collection and analysis, decision to publish, or preparation of the manuscript"

Please clarify the sources of funding (financial or material support) for your study. List the grants or organizations that supported your study, including funding received from your institution.

State what role the funders took in the study. If the funders had no role in your study, please state: “The funders had no role in study design, data collection and analysis, decision to publish, or preparation of the manuscript.”

If any authors received a salary from any of your funders, please state which authors and which funders.

If you did not receive any funding for this study, please state: “The authors received no specific funding for this work.”

Please include your amended statements within your cover letter; we will change the online submission form on your behalf.”

Response: The source of funding and role of funders were declared in the online submission system. We also included the amended statements in our cover letter for the revised manuscript as indicated.

Comment: “6. We note that you have indicated that data from this study are available upon request. PLOS only allows data to be available upon request if there are legal or ethical restrictions on sharing data publicly. For information on unacceptable data access restrictions, please see http://journals.plos.org/plosone/s/data-availability#loc-unacceptable-data-access-restrictions.

We will update your Data Availability statement on your behalf to reflect the information you provide.”

Response: In our revised cover letter, relevant data are available within the manuscript as well as in the Supporting Information files (S2 and S3 tables)

Responses to the comments by Reviewer 1:

Comment: “The article is relevant to only those practicing in Yemen as these findings cannot be generalized to other countries and populations. Hence publishing these results in local journals for local information and dissipation of knowledge would be much more useful.”

Response: We thank the Reviewer. However, we believe that publishing data about the co-infection between malaria (as a disease of poverty) and dengue (as a neglected tropical disease) would be of interest to a large audience of those interested in public health and VBDs. These two mosquito-borne diseases pose a major health problem to many countries with co-endemicity over the tropics and sub-tropics. The escalating outbreaks of dengue in many malaria-endemic countries can be a major concern beyond local interests because of mobility-driven changing epidemiology. The importance of assessing co-infection in Yemen, which is undergoing one of the worst humanitarian crises, can be relevant to other settings affected by crises and complex emergencies. This study can highlight the status of such preventable health problems in such complex emergencies. 

Comment: “2. The choice of only OPD patients with co-infection defeats the purpose of the study as the fact that dengue or malaria can be missed and thus result in complications to the patient is what is important in such co-infection.”

Response: The objective of the study was to determine mono- and co-infection with either type of infection among febrile patients. Therefore, the infection rates of malaria, dengue and malaria-dengue coinfection were determined. However, to address this important concern, we added a recommendation to conduct longitudinal studies to follow up patients with mono- or co-infection for complications (Lines 358-360).

Comment: “3. The final outcomes of the patients has not been analysed which could add a lot of value to the study.”

Response: This was a cross-sectional study with the aim to determine the proportions of malaria and dengue, as mono- or co-infection, among febrile patients in a malaria-endemic area witnessing dengue outbreaks as its outcome in relation to their clinical characteristics. With the upsurge in acute febrile illnesses that share similar clinical characteristics with malaria, the findings of this study will inform healthcare professionals about the epidemiologic situation of malaria and dengue in the study area. A recommendation about the follow up the outcomes or complications through longitudinal studies was added (Lines 358-360) because the cross-sectional nature of the present study does not permit such follow-up.

Comment: “4. The RDTs can show false positive tests in dengue when malaria is positive, and an IgM Dengue ELISA to confirm these would be more useful while generalizing the results to other non-resource limited countries / institutions.”

Response: We thank the Reviewer for the critical comment and agree with what has been suggested. For this reason, we had already highlighted this as one of the study limitations (Lines 361-367: “Although dengue was diagnosed using RDTs …………………………….… in such a resource-limited country”). We discussed the utility of RDTs that combine NS1 antigen with IgM and IgG antibodies for dengue diagnosis like those used in the study (Lines 250-254: “NS1 helps detect recent infections …………………………….…….. has been evidenced among febrile patients [41]”). We also referred in our original manuscript to the fact that irrespective of sensitivity, RDTs were able to reveal the large difference between dengue mono-infection (among about a third of febrile patients) and co-infection with malaria (among <5.0% of febrile patients), and this is a major finding highlighting the low proportion of co-infection in the study area (Lines 254-257: “Given that the RDT used in the present study might not be sensitive enough …………could be still informative”). Moreover, to avoid misinterpretation of our findings compared to those reported by ELISA or PCR elsewhere, we referred to such possible difference in our original manuscript (Lines 282–284: “Therefore, the differences in detection methods ………………… with those reported in these studies”). Following the Reviewer’s suggestion, we included a recommendation to use IgM ELISA as well as molecular techniques (Lines 364-367).

Responses to the comments by Reviewer 2:

Comment: “The authors have carried out a systematic study to assess the co-infections of malaria and dengue in Hodeidah city during the malaria transmission season (September 2018 – February 2019). The study has been conducted systematically. However, the interpretation and discussion of the study is incomplete. Recently literature is also not appropriately cited. Based on these points, the following suggestions are made to improve this study:

(i) There is a biased sex ratio of 63% males in the recruited patients in this study. This should be appropriately addressed in the discussion and the possible effect of this bias on the outcomes of this study should be discussed.”

Response: We thank the Reviewer for the comments. We revised the discussion and cited literature carefully, but if there are any other specific issues that need to be addressed or corrected, we will be pleased to address. Following the Reviewer’s comment, we have included in the discussion the issue of having recruited more men, which should be considered in further studies (Lines 303-307). However, we do not have any hypothesis to believe that co-infection may be different between men and women. This is why we did not consider it in the sample size calculation. In the revised manuscript, we added a comparison between co-infection and either type of mono-infection in relation to sociodemographic characteristics (Lines: 39-41 of the abstract, 203-206 and 297-301 of the revised manuscript as well as Table 3). We also made changes whenever necessary throughout the manuscript.

Comment: “(ii) The study has used RDT, and microscopy as methods of diagnosis for malaria. This is a partially correct methodology since ample evidence exists in literature that RDT and microscopy are not adequate for the detection of malaria- and co-infections are especially hard to detect. There should be a detailed discussion on this topic with an emphasis on recently published reports such as Mandage R, et al. Emerg Infect Dis. 2020; Kaur, C., et. al., BMC Res Notes 2020; Watson OJ, et al. BMJ Glob Health. 2019; Anstey NM, Grigg MJ. J Infect Dis. 2019; Berhane A et. al., Emerg Infect Dis. 2018; to name a few.”

Response: We agree with the Reviewer that PCR is the best alternative to detect malaria parasites in mono- and co-infections. As per the Reviewer recommendation, we discussed this in the revised manuscript and stressed on the need for cautious interpretation of the study findings, citing the indicated references by the Reviewer (Lines 244-249). Because the study area is still in the control phase of malaria, the use of microscopy and RDTs could be feasible in such a resource-limited setting. We are particularly thankful to the reviewer for the publications provided. 

Comment: “(iii) Since most of the data here is generated based only on RDT and microscopy, there is a likelihood of underrepresentation of co-infections and other species of malaria. This is incompletely discussed from lines 218-226. This should be further discussed to include all possible reasons for the underdiagnosis of malaria and dengue including but not limited to seasonality, the nature of tests used, the relative sensitivity and specificities of these tests etc.”

Response: We discussed the reasons for the possible underestimation introduced by the nature of the tests used and other factors in the revised manuscript (Lines: 262-273). 

Comment: “(iv) The authors have mentioned the demographics on unemployment status and crowding in the results section (line 167). The impact of these parameters on malaria and dengue infection/ transmission should be discussed since this has been mentioned in the result.”

Response: We analyzed and compared mono- and co-infection rates with malaria and dengue among febrile patients in relation to sociodemographic factors but did not find a statistically significant difference. As mentioned in our response above, a comparison between co-infection and either type of mono-infection in relation to sociodemographic characteristics, including the unemployment and large household size, was added (Lines: 203-206 and 299-307 as well as Table 3). 

Comment: “(v) The purpose of Fig. 1 is unclear- including a discussion on the geography of the Hodeidah city and its impact on malaria and dengue transmission and incidence, might warrant inclusion of this figure. Otherwise, it may be removed.”

Response: We preferred to delete this figure as indicated by the Reviewer.

Comment: “Minor comment: Spelling error in line 91.”

Response: Thank you. We corrected the spelling error. The misspelled word was replaced with “Subjects”.

We hope that we addressed the comments raised by the Reviewers that contributed to the improvement of the quality of our manuscript. We hope that our revised manuscript is accepted for publication in PLOS ONE, and we are pleased to receive any further comments or suggestions. 

Best regards,

Rashad Abdul-Ghani, PhD 

The Corresponding Author

---

## [Decision Letter · Decision Letter 1]

11 May 2021

PONE-D-21-03805R1

Malaria and dengue in Hodeidah city, Yemen: one-third of febrile outpatients with dengue or malaria but low proportion co-infected

PLOS ONE

Dear Dr. Abdul-Ghani,

Thank you for submitting your manuscript to PLoS ONE. After careful consideration, we feel that your manuscript will likely be suitable for publication if the authors revise it to address critical points raised now by the reviewer.  According to reviewer, there are some specific areas where further improvements would be of substantial benefit to the readers.   For your guidance, a copy of the reviewers' comments was included below.  

We look forward to receiving your revised manuscript.

Kind regards,

Luzia Helena Carvalho, Ph.D.

Academic Editor

PLOS ONE

Journal Requirements:

Reviewers' comments:

Reviewer's Responses to Questions

**Comments to the Author**

1. If the authors have adequately addressed your comments raised in a previous round of review and you feel that this manuscript is now acceptable for publication, you may indicate that here to bypass the “Comments to the Author” section, enter your conflict of interest statement in the “Confidential to Editor” section, and submit your "Accept" recommendation.

Reviewer #1: All comments have been addressed

Reviewer #2: All comments have been addressed

2. Is the manuscript technically sound, and do the data support the conclusions?

Reviewer #1: Yes

Reviewer #2: Yes

3. Has the statistical analysis been performed appropriately and rigorously? 

Reviewer #1: N/A

Reviewer #2: Yes

4. Have the authors made all data underlying the findings in their manuscript fully available?

Reviewer #1: No

Reviewer #2: Yes

5. Is the manuscript presented in an intelligible fashion and written in standard English?

Reviewer #1: Yes

Reviewer #2: Yes

6. Review Comments to the Author

Reviewer #1: The authors have addressed all the queries as brought about in the review. The article may be accepted

Reviewer #2: Authors responded to most comments and incorporated most of the suggestions. While the major issue with underestimation of infection frequencies still exists, in most settings, RDT and microscopy are the methods used for evaluation of malaria and dengue co-infections. A short discussion on other potential co-infections that may be present but have not been investigated in the current study, may be included.

7. PLOS authors have the option to publish the peer review history of their article (what does this mean?). If published, this will include your full peer review and any attached files.

Reviewer #1: **Yes: **Soundarya Mahalingam

Reviewer #2: No

---

## [Author Response · Author response to Decision Letter 1]

22 May 2021

Dear Editor,

Thank you for giving us this second opportunity to further improve the quality of our manuscript. Our thanks are also due to the Reviewers for their helpful comments and concerns. The manuscript has been revised after considering these comments and carefully adhering to the editorial requirements of PLOS ONE. Changes in the revised manuscript are made with track changes. The line numbers mentioned below refer to those in the revised manuscript with track changes. Detailed responses to the comments raised are as follows:

Responses to the comments by the Academic Editor:

Comment: “Thank you for submitting your manuscript to PLoS ONE. After careful consideration, we feel that your manuscript will likely be suitable for publication if the authors revise it to address critical points raised now by the reviewer. According to reviewer, there are some specific areas where further improvements would be of substantial benefit to the readers. For your guidance, a copy of the reviewers' comments was included below”

Response: We thank the Academic Editor and the Reviewers for the helpful comments and careful consideration to improve the quality of our manuscript. We addressed the additional concerns and issues raised by Reviewer 2 in the revised version of the manuscript. Because World Malaria Report is issued annually to provide statistics on malaria during the previous year, we updated the figures on malaria epidemiology in the revised manuscript as per the latest edition of the report issued in 2020 instead of those provided in the 2019 edition (Lines 57& 58 and Lines 69 & 70 of the revised manuscript) and updated citation #1 in the list of references, accordingly. In addition, we proofread the manuscript for language issues and fixed the minor issues we found (marked with track changes in the revised manuscript).

Responses to PLOS ONE requirements:

Comment: “Please review your reference list to ensure that it is complete and correct. If you have cited papers that have been retracted, please include the rationale for doing so in the manuscript text, or remove these references and replace them with relevant current references. Any changes to the reference list should be mentioned in the rebuttal letter that accompanies your revised manuscript. If you need to cite a retracted article, indicate the article’s retracted status in the References list and also include a citation and full reference for the retraction notice.”

Response: We double-checked the reference list of our manuscript, which was originally created using EndNote software, for its completeness, correctness and formatting according to PLOS ONE’s style after removing the EndNote links. We did not cite retracted papers in our manuscript. We added one citation to our revised manuscript (#66; Prasad et al.) to enrich the discussion added in response to the comment by Reviewer 2.

Responses to the comment by Reviewer 1:

Comment: “The authors have addressed all the queries as brought about in the review. The article may be accepted.”

Response: We thank the Reviewer for the supportive feedback. We just noted that the Reviewer answered with “No” to question #4 “Have the authors made all data underlying the findings in their manuscript fully available?”, but we confirm that we made all data fully available and uploaded an Excel data file with the first version of our revised manuscript submitted to the journal.

Responses to the comment by Reviewer 2:

Comment: “Authors responded to most comments and incorporated most of the suggestions. While the major issue with underestimation of infection frequencies still exists, in most settings, RDT and microscopy are the methods used for evaluation of malaria and dengue co-infections. A short discussion on other potential co-infections that may be present but have not been investigated in the current study, may be included.”

Response: We thank the Reviewer for this important comment. In the first version of our revised manuscript, we elaborated on discussing the issue with the underestimation of mono- and co-infection with either type of infection based on the diagnosis of malaria with microscopy and RDTs and dengue with RDTs in light of the Reviewer’s comments on our original submission (lines 244–272: “Malaria diagnosis in the present study using microscopy and antigen detection using RDTs may underestimate….. preferably as a single format, for both types of infection has been suggested [14].” Following the suggestion above, we added a short discussion on other potential co-infections that may be present but have not been investigated in the present study (Lines 350–369 of the revised manuscript). We also added a sentence about this important issue to the recommendations of the study (Lines 396–398) to highlight its importance.

We hope that we addressed the comments raised by the Editor and Reviewers that contributed to further improving the quality of our manuscript. We also hope that our revised manuscript is accepted for publication in PLOS ONE, and we are pleased to receive any further comments or suggestions. 

Best regards,

Rashad Abdul-Ghani, PhD 

The Corresponding Author

---

## [Decision Letter · Decision Letter 2]

8 Jun 2021

Malaria and dengue in Hodeidah city, Yemen: one-third of febrile outpatients with dengue or malaria but low proportion co-infected

PONE-D-21-03805R2

Dear Dr. Abdul-Ghani,

We’re pleased to inform you that your manuscript has been judged scientifically suitable for publication and will be formally accepted for publication once it meets all outstanding technical requirements.

Kind regards,

Luzia Helena Carvalho, Ph.D.

Academic Editor

PLOS ONE

Additional Editor Comments (optional):

Reviewers' comments:

Reviewer's Responses to Questions

**Comments to the Author**

1. If the authors have adequately addressed your comments raised in a previous round of review and you feel that this manuscript is now acceptable for publication, you may indicate that here to bypass the “Comments to the Author” section, enter your conflict of interest statement in the “Confidential to Editor” section, and submit your "Accept" recommendation.

Reviewer #1: All comments have been addressed

Reviewer #2: All comments have been addressed

2. Is the manuscript technically sound, and do the data support the conclusions?

Reviewer #1: Yes

Reviewer #2: Yes

3. Has the statistical analysis been performed appropriately and rigorously? 

Reviewer #1: Yes

Reviewer #2: Yes

4. Have the authors made all data underlying the findings in their manuscript fully available?

Reviewer #1: No

Reviewer #2: Yes

5. Is the manuscript presented in an intelligible fashion and written in standard English?

Reviewer #1: Yes

Reviewer #2: Yes

6. Review Comments to the Author

Reviewer #1: Manuscript is written keeping in mind all the necessary requirements for the journal. All previous comments have been addressed in the revision

Reviewer #2: All the comments have been addressed in the manuscript. It now appears well-balanced article that discusses the important issue of co-infections of multiple co-seasonal and co-endemic pathogens.

7. PLOS authors have the option to publish the peer review history of their article (what does this mean?). If published, this will include your full peer review and any attached files.

Reviewer #1: No

Reviewer #2: No

---

## [Editor Report · Acceptance letter]

17 Jun 2021

PONE-D-21-03805R2 

Malaria and dengue in Hodeidah city, Yemen: high proportion of febrile outpatients with dengue or malaria, but low proportion co-infected 

Dear Dr. Abdul-Ghani:

I'm pleased to inform you that your manuscript has been deemed suitable for publication in PLOS ONE. Congratulations! Your manuscript is now with our production department. 

Kind regards, 

on behalf of

Dr. Luzia Helena Carvalho 

Academic Editor

PLOS ONE